# Association between handgrip strength, handgrip strength asymmetry, and anxiety in Korean older adults: The Korean National Health and Nutrition Examination Survey 2022

**Sang-Youn Choi[1], Su-Min Park[2,3]\*, Eun-Cheol Park[3,4]\***

**1** Medical Course, Yonsei University College of Medicine, Seoul, Republic of Korea, **2** Department of Public Health, Graduate School, Yonsei University, Seoul, Republic of Korea, **3** Institute of Health Services Research, Yonsei University, Seoul, Republic of Korea, **4** Department of Preventive Medicine, Yonsei University College of Medicine, Seoul, Republic of Korea

\* gemmaa3737@yuhs.ac (S-MP); ECPARK@yuhs.ac (E-CP)

## Abstract

Low handgrip strength (HGS) and HGS asymmetry are associated with age-related physical and mental disorders in older adults. This study aimed to examine the association between HGS-related factors and anxiety to evaluate whether HGS assessments can assist in identifying anxiety risk. In total, 1,750 participants from the Korea National Health and Nutrition Examination Survey of 2022 were included in this study. Individuals whose HGS values were below the 20th percentile of the study population stratified by sex were classified into the low-HGS group. Anxiety was assessed using the generalized anxiety disorder with a 7-item scale. Multiple logistic regression was used to analyze the relationship between HGS level and asymmetry and anxiety, adjusting for covariates. Overall, 70 (8.7%) men and 123 (13.0%) women had anxiety. Elevated odds of anxiety were observed in older women with low HGS (adjusted odds ratio: 2.17, 95% confidence interval: 1.31–3.61). There was a positive correlation between the degree of asymmetrical HGS and anxiety among women. This study found positive associations between low HGS, HGS asymmetry, and anxiety in older Korean women. This population may require specific interventions to help maintain good mental health.

## Introduction

Anxiety is more common in older adults than in young people [1], and its prevalence is increasing. For example, the rates of anxiety among the United States adults increased from 5.12% in 2008 to 6.68% in 2018 [2]. Meanwhile, anxiety among Brazilian adults showed a sharp increase during the COVID-19 pandemic [3].

Populations in which 14% and 20% of people are older than 65 years are considered an "aged" and a "super-aged" society, respectively. Korea became an aged society in 2018 and is expected to become a super-aged society by 2025, as there are currently more than

**Data availability statement:** All KNHANES files are available from the KDCA database (URLs https://knhanes.kdca.go.kr/knhanes/sub03/sub03_02_05.do).

**Funding:** The author(s) received no specific funding for this work.

**Competing interests:** The authors have declared that no competing interests exist.

10 million people older than 65 years [4]. The prevalence of anxiety among older adults in Korea was 11.0%, according to the Korea National Health and Nutrition Examination Survey (KNHANES) in 2022.

Anxiety in older adults is associated with increased morbidity and mortality rates [5–7]. Older adults with anxiety are at higher risk of cardiovascular disease and cognitive decline [8], necessitating specific interventions. However, anxiety symptoms in older adults are often underestimated because mental health changes tend to be attributed to physical diseases in this population [8]. Therefore, novel approaches to screening, diagnosing, and managing anxiety in older adults are required.

We hypothesized that sarcopenia may be a good indicator of anxiety. Sarcopenia is associated with many aging-associated diseases, such as osteoporosis, type 2 diabetes mellitus, and cardiovascular disease [9–11]. Moreover, it has been reported to contribute to depression, which is also a mental disorder related to anxiety [12,13]. Therefore, sarcopenia may be associated with anxiety.

Measuring handgrip strength (HGS) is one way to diagnose sarcopenia [14]. Hence, HGS might be used as an assistant index for diagnosing anxiety if the association between the two factors is shown. Sarcopenia and low HGS are associated with aging-related diseases and depression [15–18]. Measuring HGS involves using portable devices, which makes it an accessible indicator.

Asymmetrical HGS may also be associated with anxiety. HGS difference between hands ranging from 0% to 10% is considered normal [19]. In contrast, an asymmetry of > 10% is associated with depression and age-related disease [20–22].

Some studies have found an association between HGS and age-related or mental diseases in older adults. In addition, a study of Polish older adults found that better HGS is related to a lower frequency of pain and anxiety, which was investigated using the EuroQol-5D questionnaire [23]. However, no study to date has examined the relationship between HGS-related factors and anxiety in older Koreans using the Generalized Anxiety Disorder 7-item (GAD-7), which is more specific for measuring anxiety than the EuroQol-5D questionnaire. Therefore, in this study, we aimed to explore the association between anxiety, low HGS, and HGS asymmetry in Korean older adults.

## Methods

The data analyzed in this study were obtained from the 2022 KNHANES IX. KNHANES is a nationwide cross-sectional survey conducted by the Korea Disease Control and Prevention Agency. It assesses the health and nutritional status of Koreans, capturing variables such as obesity status, blood pressure values, and diabetes diagnosis. It is used to inform national policies and for global comparisons of health outcomes.

### Study population

The GAD-7 was first included in the 2021 KNHANES. However, as the HGS was not investigated in 2021 and the survey for 2023 was not available at the time of writing, this study only used data from 2022. In total, 6,265 individuals were included in the data for 2022. A total of 4,023 individuals under the age of 60 years, and 492 individuals lacking values for study variables were excluded. Therefore, 1,750 individuals were included in the analysis.

### Variables

We used the GAD-7 to select participants with anxiety. The GAD-7 comprises seven questions: participants answered each question on a scale from 0 to 3, with 0 indicating no anxiety

at all and 3 indicating anxiety nearly every day. The answers to each question are summed to obtain a final score that ranges from 0 to 21. The score 0–4 indicated no anxiety, 5–9 mild anxiety, 10–14 moderate anxiety, and 15–21 severe anxiety [24]. We used a threshold of 5; therefore, those with a score of 4 or lower were classified as not having anxiety and those with a score of 5 or higher were classified as having anxiety.

The HGS of each hand was measured twice using a digital dynamometer. Individuals with defects of the arm/hand/thumb, defects or fractures of fingers other than the thumb, paralysis of the hand, or cast or bandage of the hand/wrist were excluded from the study. The participants gripped the dynamometer for 3 s with their maximum strength in the upright position. There was a break of 60 s between the two measurements. The strongest measurements were obtained from the dominant hand. Following the criteria suggested by the Asian Working Group of Sarcopenia, we used the 20th percentile of the study population's HGS, which was stratified by sex, as a threshold: high for 20–100th percentile, and low for 0-20th percentile [25]. For additional analyses, the 10th percentile was used to divide the "low" group for comparisons between "low" and "very low" HGS. HGS asymmetry between two hands lower than 10% is considered normal. To investigate the association between the degree of asymmetry and anxiety, participants were categorized into three groups: symmetry for those with 0–10% asymmetry, moderate asymmetry for those with 10–20% asymmetry, and prominent asymmetry for those with over 20% asymmetry.

The data were stratified according to sex. Age (60–69, 70 years, or older), educational level (high school or lower, college or higher), working status (working or not working), area of residence (metropolitan or rural), and marital status (with or without a spouse) were included as demographic and socioeconomic factors. Obesity (body mass index $\geq 25$ kg/m², between 18.5 and 25 kg/m², $< 18.5$ kg/m²) [26], frequency of drinking (less than once a year, once a year or more), sleep duration (less than 6 h, more than 6 h per night), stress level (a lot, a little), physical activity (sufficient, insufficient), and lifetime smoking history (yes, no) were included as health-related factors. "Sufficient" or "insufficient" physical activity was defined by Aerobic Physical Activity Prevalence guidelines [27]. "Sufficient" physical activity was defined as medium-intensity physical activity of more than 2.5 h per week or high-intensity physical activity of more than 1 h and 15 min per week, or both forms of activity at a total of $\geq 2.5$ h per week, using a conversion of 1 min of high-intensity physical activity as equal to 2 min of medium-intensity physical activity. High-intensity physical activities included lifting and carrying heavy objects, digging, running, and swimming. Medium-intensity physical activities included lifting and carrying light objects, golf and jogging.

## Statistical analysis

The Chi-square tests were performed to compare participant characteristics. We used stratification, clustering, and weighting variables in the KNHANES data to ensure population representativeness. We used weighting variables that consider both health and nutrition factors. Age, education level, working state, residence, marital status, obesity, drinking, sleep duration, stress level, physical activity, and smoking were used as confounders. These variables were included solely as control variables and not as independent predictors of anxiety. Multiple logistic regression was used to investigate the association between HGS-related factors (low HGS and HGS asymmetry) and anxiety after considering confounding variables. Subgroup analysis stratified by each independent variable was conducted to examine the independent effects of the variables and determine any group with a high association between low HGS and anxiety. Finally, we calculated the odds ratio (OR) of anxiety in the groups with low HGS and HGS asymmetry using multiple logistic regression. For all logistic regression analyses, ORs and 95% confidence intervals (CIs) were calculated. The variance inflation factors for the

variables were smaller than 1.15, indicating that there was no problem with multicollinearity. All analyses were performed using SAS software (version 9.4; SAS Institute, Cary, North Carolina, USA), and p-values < 0.05 were considered statistically significant.

### Ethics statement

The KCDC Research Ethics Review Committee has reviewed and approved the KNHANES annually since 2007. The KNHANES datasets are published online for research purposes. The committee operates according to the KCDC Research Ethics Review Committee's standard guidelines. All survey participants filled out a consent form before participation. As this study utilized publicly available data, additional institutional review board approval was not necessary, as per Article 2.2 of the Enforcement Rule of the Bioethics and Safety Act in Korea.

## Results

Table 1 shows the general characteristics of the study population stratified by sex. A total of 1,750 participants, including 802 (45.8%) men and 948 (54.2%) women, were included in the analysis. Among them, 70 (8.7%) men and 123 (13.0%) women experienced anxiety. As the HGS threshold was the 20th percentile, 20% of participants in each sex group were classified as having low HGS. Among participants with low HGS, 11.7% of men and 19.8% of women had anxiety. Meanwhile, among participants with high HGS, 8.0% of men and 11.2% of women had anxiety. The Chi-square results showed a statistically significant relationship between low HGS and anxiety in women, but not in men.

Table 2 shows the results of multiple logistic regression analysis between HGS and anxiety. Both before and after adjusting for covariates, the risk of having anxiety was significantly higher among women who had low HGS (crude OR [cOR]: 1.33, 95% CI: 0.67–2.63 in men, cOR: 1.989, 95% CI: 1.23–3.21 in women, adjusted OR [aOR]: 1.22, 95% CI: 0.62–2.43 in men, aOR: 2.17, 95% CI: 1.31–3.61 in women).

The results of the subgroup analyses are presented in Tables 3 and 4. Both before and after adjusting for covariates, no statistically significant differences were found in any male subgroup, in contrast to female subgroups. These results show trends similar to those of the main results (Table 2). The female subgroups with a significantly higher aOR for "having anxiety" and "having low HGS" included those aged 70 years and older, with a high school education or lower, both employed or unemployed, married, non-drinkers, those sleeping more than 6 h per night, those with high stress, both physically active and inactive individuals, and non-smokers.

After subgrouping individuals according to the 10th and 20th HGS percentiles (very low for 0–10% HGS, low for 10–20% HGS, and high for > 20% HGS), we conducted multiple logistic regression analysis using the high group as a reference. To examine the association between HGS asymmetry and anxiety, participants were categorized into three groups by a threshold of 10% and 20% asymmetry, and multiple logistic regression analyses were performed using the symmetry group as a reference. The results of these two analyses are presented in Table 5. No effects were observed among men. However, among women, the very low HGS group had significantly higher odds of anxiety compared to the high HGS group (aOR: 3.22, 95% CI: 1.67–6.20). There was also a positive correlation between lower HGS and higher odds of anxiety (aOR for low HGS group: 1.46, aOR for very low HGS group: 3.22). A similar correlation was observed in women's HGS asymmetry: The more severe the HGS asymmetry, the higher the odds of anxiety (aOR for the moderate HGS asymmetry group: 1.37; aOR for the prominent HGS asymmetry group: 1.83).

**Table 1. Socioeconomic and health-related characteristics of all study participants according to anxiety.**

| Variables | Anxiety | | | | | | | | | |
|---|---|---|---|---|---|---|---|---|---|---|
| | Male (N = 802) | | | | | Female (N = 948) | | | | |
| | NO | | YES | | p-value | NO | | YES | | p-value |
| | N | (%) | N | (%) | | N | (%) | N | (%) | |
| **Handgrip strength** | | | | | 0.1299 | | | | | 0.0016 |
| High | 589 | (92.0) | 51 | (8.0) | | 671 | (88.8) | 85 | (11.2) | |
| Low | 143 | (88.3) | 19 | (11.7) | | 154 | (80.2) | 38 | (19.8) | |
| **Handgrip strength symmetry** | | | | | 0.9298 | | | | | 0.2985 |
| Symmetric | 464 | (91.3) | 44 | (8.7) | | 477 | (88.0) | 65 | (12.0) | |
| Asymmetry | 268 | (91.2) | 26 | (8.8) | | 348 | (85.7) | 58 | (14.3) | |
| **Age** | | | | | 0.981 | | | | | 0.5165 |
| 60-69 | 388 | (91.3) | 37 | (8.7) | | 464 | (86.4) | 73 | (13.6) | |
| 70- | 344 | (91.2) | 33 | (8.8) | | 361 | (87.8) | 50 | (12.2) | |
| **Education Level** | | | | | 0.0095 | | | | | 0.1292 |
| High school or lower | 534 | (89.7) | 61 | (10.3) | | 736 | (87.6) | 104 | (12.4) | |
| College or higher | 198 | (95.7) | 9 | (4.3) | | 89 | (82.4) | 19 | (17.6) | |
| **Working state** | | | | | 0.672 | | | | | 0.521 |
| No | 354 | (91.7) | 32 | (8.3) | | 488 | (87.6) | 69 | (12.4) | |
| Yes | 378 | (90.9) | 38 | (9.1) | | 337 | (86.2) | 54 | (13.8) | |
| **Residence** | | | | | 0.8969 | | | | | 0.0558 |
| Metropolitan | 528 | (91.2) | 51 | (8.8) | | 611 | (88.3) | 81 | (11.7) | |
| Rural | 204 | (91.5) | 19 | (8.5) | | 214 | (83.6) | 42 | (16.4) | |
| **Marital status** | | | | | 0.374 | | | | | 0.363 |
| No | 79 | (88.8) | 10 | (11.2) | | 276 | (83.9) | 53 | (16.1) | |
| Yes | 653 | (91.6) | 60 | (8.4) | | 549 | (88.7) | 70 | (11.3) | |
| **Obesity** | | | | | 0.9395 | | | | | 0.0108 |
| Underweight | 18 | (90.0) | 2 | (10.0) | | 16 | (66.7) | 8 | (33.3) | |
| Normal | 464 | (91.5) | 43 | (8.5) | | 508 | (87.4) | 73 | (12.6) | |
| Obese | 250 | (90.9) | 25 | (9.1) | | 301 | (87.8) | 42 | (12.2) | |
| **Drink** | | | | | 0.1314 | | | | | 0.153 |
| No | 209 | (88.9) | 26 | (11.1) | | 460 | (85.7) | 77 | (14.3) | |
| Yes | 523 | (92.2) | 44 | (7.8) | | 365 | (88.8) | 46 | (11.2) | |
| **Sleep duration** | | | | | 0.4911 | | | | | 0.0212 |
| More than 6h | 500 | (91.7) | 45 | (8.3) | | 493 | (89.2) | 60 | (10.8) | |
| Less than 6h | 232 | (90.3) | 25 | (9.7) | | 332 | (84.1) | 63 | (15.9) | |
| **Stress** | | | | | <0.001 | | | | | <0.001 |
| Low | 241 | (99.6) | 1 | (0.4) | | 240 | (97.6) | 6 | (2.4) | |
| High | 491 | (87.7) | 69 | (12.3) | | 585 | (83.3) | 117 | (16.7) | |
| **Physical Activity** | | | | | 0.6694 | | | | | 0.7343 |
| Insufficient | 441 | (90.9) | 44 | (9.1) | | 556 | (87.3) | 81 | (12.7) | |
| Sufficient | 291 | (91.8) | 26 | (8.2) | | 269 | (86.5) | 42 | (13.5) | |
| **Smoke** | | | | | 0.6592 | | | | | 0.009 |
| Non-smoker | 558 | (91.0) | 55 | (9.0) | | 806 | (87.5) | 115 | (12.5) | |
| Smoker | 174 | (92.1) | 15 | (7.9) | | 19 | (70.4) | 8 | (29.6) | |
| **Total** | 732 | (91.3) | 70 | (8.7) | | 825 | (87.0) | 123 | (13.0) | |

Variables are presented as numbers and percentages.

**Table 2.  Association between handgrip strength and anxiety.**

| Variables | | Anxiety (unadjusted) | | | Anxiety (adjusted) | | |
|---|---|---|---|---|---|---|---|
| | | cOR | 95% CI | p-value | aOR | 95% CI | p-value |
| **Male** | **HGS** | | | 0.4102 | | | 0.5654 |
| | High | 1 | | | 1 | | |
| | Low | 1.33 | (0.67–2.63) | | 1.221 | (0.62–2.43) | |
| **Female** | **HGS** | | | 0.0050 | | | 0.0029 |
| | High | 1 | | | 1 | | |
| | Low | 1.989 | (1.23–3.21) | | 2.173 | (1.31–3.61) | |

Table 6 shows the results of multiple logistic regression analysis for the effects of HGS and HGS asymmetry on anxiety. No effect was observed among men. Among women, the group with "low HGS and moderate asymmetry" (aOR: 2.98, 95% CI: 1.49–5.94) and that with "low HGS and prominent asymmetry" (aOR: 3.16, 95% CI: 1.37–7.29) had higher odds of having anxiety. The odds of anxiety increased as the degree of asymmetry increased and HGS became lower among women.

## Discussion

In this study, we found that a low HGS was significantly associated with a higher odds of anxiety among older Korean women. Furthermore, we found a positive correlation between the degree of HGS asymmetry and risk of anxiety in women.

A previous study showed that high HGS was related to a lower prevalence of pain and anxiety in older Polish adults [23]. Our results generally agree with those of this previous study. However, our results were different from those of previous studies, showing significant effects among women but not among men. Population-level and methodological differences may account for these discrepancies.

Another previous study showed that the weaker the HGS in older Korean women, the higher the risk of depression, while no significant association was found in men [28]. A previous study has shown an association between anxiety and depression [29,30]. Combining these two results, we can infer that there is an association between low HGS and anxiety, specifically in women. The results of this study are consistent with this inference.

Although the mechanism by which low HGS is associated with anxiety remains unclear, we proposed the following explanation: older individuals with low HGS were more likely to have poor physical function. A low HGS is associated with upper limb disability in women with bilateral idiopathic carpal tunnel syndrome [31]. Older people with low physical function may be unable to get out of their beds, restricting their ability to engage in the activities of daily living, which would reduce their quality of life. Low HGS is associated with poor quality of life in cancer survivors [32]. Poor quality of life and functional impairment can lead to anxiety [33]. Anxiety may also contribute to a low HGS. Individuals with anxiety are more likely to experience sleep deprivation, which induces muscle loss and impairs regeneration [34,35].

Women appeared to have a greater inclination to experience anxiety due to low HGS compared to men. The 20th percentile HGS threshold for HGS in men (30.0 kg) is the 95th percentile of HGS in women. As low HGS for a man is high HGS for a woman, a man with low HGS may retain sufficient power to engage in daily activities, which would reduce his risk of anxiety. Men may also have a greater margin for HGS decline before experiencing the effects of weakness, compared to women. In addition, women are more likely to experience

**Table 3. Subgroup analysis of the relationship between handgrip strength and anxiety among male.**

| Variables | Anxiety | | | | | | | |
| --- | --- | --- | --- | --- | --- | --- | --- | --- |
| | Unadjusted | | | | Adjusted | | | |
| | High HGS | Low HGS | | | High HGS | Low HGS | | |
| | cOR | cOR | 95% CI | p-value | aOR | aOR | 95% CI | p-value |
| **Age** | | | | | | | | |
| 60-69 | 1 | 0.83 | (0.24–2.95) | 0.7749 | 1 | 0.75 | (0.19–2.88) | 0.6704 |
| 70- | 1 | 2.02 | (0.88–4.62) | 0.0950 | 1 | 1.72 | (0.68–4.34) | 0.2515 |
| **Education Level** | | | | | | | | |
| High school or lower | 1 | 1.35 | (0.69–2.64) | 0.3731 | 1 | 1.58 | (0.78–3.12) | 0.2101 |
| College or higher | 1 | – | – | <0.001 | 1 | – | – | <0.001 |
| **Working state** | | | | | | | | |
| No | 1 | 1.54 | (0.60–3.98) | 0.3715 | 1 | 1.24 | (0.48–3.24) | 0.6589 |
| Yes | 1 | 1.26 | (0.43–3.67) | 0.6688 | 1 | 0.86 | (0.28–2.63) | 0.7834 |
| **Residence** | | | | | | | | |
| Metropolitan | 1 | 0.74 | (0.32–1.73) | 0.4857 | 1 | 0.75 | (0.31–1.80) | 0.5138 |
| Rural | 1 | 5.11 | (1.63–15.97) | 0.0054 | 1 | 3.20 | (1.19–8.57) | 0.0213 |
| **Marital status** | | | | | | | | |
| No | 1 | 0.72 | (0.12–4.40) | 0.7154 | 1 | 0.34 | (0.03–3.77) | 0.3739 |
| Yes | 1 | 1.45 | (0.69–3.06) | 0.3254 | 1 | 1.42 | (0.66–3.07) | 0.3717 |
| **Obesity** | | | | | | | | |
| Underweight | 1 | 1.10 | (0.05–23.86) | 0.9503 | 1 | – | – | <0.001 |
| Normal | 1 | 1.63 | (0.75–3.55) | 0.2199 | 1 | 1.63 | (0.72–3.71) | 0.2381 |
| Obese | 1 | 0.77 | (0.16–3.81) | 0.7506 | 1 | 0.64 | (0.13–3.20) | 0.5482 |
| **Drink** | | | | | | | | |
| No | 1 | 1.85 | (0.59–5.81) | 0.2870 | 1 | 1.58 | (0.45–5.50) | 0.4704 |
| Yes | 1 | 0.95 | (0.40–2.24) | 0.9084 | 1 | 1.05 | (0.43–2.61) | 0.9100 |
| **Sleep duration** | | | | | | | | |
| More than 6h | 1 | 1.40 | (0.59–3.31) | 0.4450 | 1 | 1.04 | (0.40–2.72) | 0.9327 |
| Less than 6h | 1 | 1.19 | (0.36–3.89) | 0.7741 | 1 | 1.23 | (0.42–3.55) | 0.7100 |
| **Stress** | | | | | | | | |
| Low | 1 | – | – | <0.001 | 1 | – | – | <0.001 |
| High | 1 | 1.25 | (0.60–2.63) | 0.5532 | 1 | 1.16 | (0.57–2.33) | 0.6821 |
| **Physical Activity** | | | | | | | | |
| Insufficient | 1 | 1.25 | (0.53–0.29) | 0.6113 | 1 | 1.08 | (0.49–2.39) | 0.8478 |
| Sufficient | 1 | 1.42 | (0.50–4.07) | 0.5125 | 1 | 1.89 | (0.58–6.21) | 0.2904 |
| **Smoke** | | | | | | | | |
| Non-smoker | 1 | 1.29 | (0.61–2.75) | 0.5001 | 1 | 1.18 | (0.54–2.56) | 0.6753 |
| Smoker | 1 | 1.36 | (0.33–5.65) | 0.6664 | 1 | 1.29 | (0.27–6.28) | 0.7489 |

anxiety than men [36,37]. Therefore, they can be easily affected by factors that cause anxiety. A patriarchal mindset prevalent among the current older Korean adults may also contribute to this difference. Having grown up in a patriarchal society, many of them hold the belief that men should earn money while women should handle housework. As a result, retired men often do not feel obligated to engage in any form of work, including domestic chores, whereas women experience pressure to manage household responsibilities. This dynamic means that women, rather than men, are more likely to feel anxious when low HGS prevents them from

**Table 4. Subgroup analysis of the relationship between handgrip strength and anxiety among female.**

| Variables | Anxiety | | | | | | | |
|---|---|---|---|---|---|---|---|---|
| | Unadjusted | | | | Adjusted | | | |
| | High HGS | Low HGS | | | High HGS | Low HGS | | |
| | cOR | cOR | 95% CI | p-value | aOR | aOR | 95% CI | p-value |
| **Age** | | | | | | | | |
| 60-69 | 1 | 2.05 | (0.98–4.27) | 0.0553 | 1 | 2.16 | (1.00–4.70) | 0.0513 |
| 70- | 1 | 2.32 | (1.11–4.85) | 0.0253 | 1 | 2.20 | (1.01–4.79) | 0.0467 |
| **Education Level** | | | | | | | | |
| High school or lower | 1 | 2.01 | (1.22–3.30) | 0.0064 | 1 | 2.06 | (1.22–3.47) | 0.0071 |
| College or higher | 1 | 1.92 | (0.45–8.18) | 0.3780 | 1 | 3.41 | (0.70–16.64) | 0.1279 |
| **Working state** | | | | | | | | |
| No | 1 | 2.00 | (1.07–3.74) | 0.0312 | 1 | 2.04 | (1.03–4.04) | 0.0419 |
| Yes | 1 | 2.39 | (0.99–5.72) | 0.0515 | 1 | 2.99 | (1.11–8.09) | 0.0312 |
| **Residence** | | | | | | | | |
| Metropolitan | 1 | 1.81 | (1.03–3.18) | 0.0387 | 1 | 1.92 | (0.98–3.78) | 0.0578 |
| Rural | 1 | 2.30 | (0.96–5.50) | 0.0610 | 1 | 2.54 | (0.92–6.98) | 0.0715 |
| **Marital status** | | | | | | | | |
| No | 1 | 1.61 | (0.79–3.28) | 0.1876 | 1 | 1.96 | (0.93–4.13) | 0.0762 |
| Yes | 1 | 2.16 | (1.02–4.58) | 0.0452 | 1 | 2.48 | (1.16–5.29) | 0.0192 |
| **Obesity** | | | | | | | | |
| Underweight | 1 | 6.72 | (0.97–46.47) | 0.0536 | 1 | – | – | 0.0010 |
| Normal | 1 | 1.67 | (0.86–3.26) | 0.1309 | 1 | 2.02 | (0.97–4.21) | 0.0596 |
| Obese | 1 | 2.00 | (0.87–4.61) | 0.1015 | 1 | 2.45 | (0.94–6.38) | 0.0668 |
| **Drink** | | | | | | | | |
| No | 1 | 1.91 | (1.01–3.62) | 0.0460 | 1 | 2.44 | (1.20–4.95) | 0.0137 |
| Yes | 1 | 2.16 | (0.93–4.99) | 0.0714 | 1 | 1.80 | (0.71–4.57) | 0.2177 |
| **Sleep duration** | | | | | | | | |
| More than 6h | 1 | 2.13 | (1.07–4.25) | 0.0316 | 1 | 2.34 | (1.15–4.76) | 0.0195 |
| Less than 6h | 1 | 1.78 | (0.88–3.58) | 0.1078 | 1 | 1.94 | (0.91–4.14) | 0.0878 |
| **Stress** | | | | | | | | |
| Low | 1 | 6.82 | (1.51–30.91) | 0.0131 | 1 | 4.83 | (0.94–24.73) | 0.0585 |
| High | 1 | 1.96 | (0.26–2.53) | 0.0123 | 1 | 2.04 | (1.19–3.50) | 0.0096 |
| **Physical Activity** | | | | | | | | |
| Insufficient | 1 | 1.91 | (1.13–3.25) | 0.0166 | 1 | 2.07 | (1.11–3.89) | 0.0233 |
| Sufficient | 1 | 2.26 | (1.00–5.10) | 0.0507 | 1 | 2.88 | (1.21–6.83) | 0.0169 |
| **Smoke** | | | | | | | | |
| Non-smoker | 1 | 2.00 | (1.23–3.25) | 0.0057 | 1 | 2.18 | (1.30–3.65) | 0.0033 |
| Smoker | 1 | 1.47 | (0.16–13.84) | 0.7326 | 1 | – | – | <0.001 |

fulfilling their expected roles. Given that 49.4% of the male survey population was not working, the disparity between retired men and women may significantly amplify the impact of low HGS on women's anxiety.

Similar to low HGS, HGS asymmetry was associated with anxiety only in women. A possible reason for this association is as follows. HGS asymmetry can contribute to functional disability [38]. Functional disabilities can lead to anxiety [33]. Therefore, HGS asymmetry may lead to anxiety. The reason men did not show significant results in the HGS asymmetry analysis might be the same as that behind the results of the low HGS analysis. As men have

**Table 5. Association between HGS-related factors and anxiety.**

| Variables | | Anxiety (unadjusted) | | | Anxiety (adjusted) | | |
|---|---|---|---|---|---|---|---|
| | | cOR | 95% CI | p-value | aOR | 95% CI | p-value |
| **Male** | **HGS** | | | | | | |
| | High | 1 | | | 1 | | |
| | Low | 1.16 | (0.44–3.03) | 0.7601 | 1.03 | (0.39–2.68) | 0.9574 |
| | Very low | 1.51 | (0.65–3.53) | 0.3378 | 1.47 | (0.62–3.53) | 0.3827 |
| | **HGS asymmetry** | | | | | | |
| | Symmetry | 1 | | | 1 | | |
| | Moderate asymmetry | 1.44 | (0.76–2.73) | 2.6390 | 1.52 | (0.75–3.07) | 0.2407 |
| | Prominent asymmetry | 1.05 | (0.42–2.67) | 0.9128 | 0.95 | (0.38–2.38) | 0.9195 |
| **Female** | **HGS** | | | | | | |
| | High | 1 | | | 1 | | |
| | Low | 1.35 | (0.64–2.84) | 0.4218 | 1.46 | (0.69–3.08) | 0.3176 |
| | Very low | 2.77 | (1.55–4.94) | <0.001 | 3.22 | (1.67–6.20) | <0.001 |
| | **HGS asymmetry** | | | | | | |
| | Symmetry | 1 | | | 1 | | |
| | Moderate asymmetry | 1.24 | (0.79–1.94) | 0.3567 | 1.37 | (0.83–2.27) | 0.2126 |
| | Prominent asymmetry | 1.78 | (0.95–3.35) | 0.0728 | 1.83 | (0.90–3.72) | 0.0948 |

**Table 6. Effects of HGS and HGS asymmetry on anxiety.**

| Variables | | | Anxiety | | | | | |
|---|---|---|---|---|---|---|---|---|
| | | | High HGS | | | Low HGS | | |
| | | | OR | 95% CI | p-value | OR | 95% CI | p-value |
| **Male** | **Unadjusted** | Symmetry | 1 | | | 1.29 | (0.52–3.20) | 0.5857 |
| | | Moderate asymmetry | 1.52 | (0.70–3.29) | 0.2923 | 1.49 | (0.55–4.06) | 0.4302 |
| | | Prominent asymmetry | 0.68 | (0.15–3.14) | 0.6141 | 2.08 | (0.59–7.31) | 0.2528 |
| | **Adjusted** | Symmetry | 1 | | | 1.16 | (0.45–3.00) | 0.7618 |
| | | Moderate asymmetry | 1.59 | (0.70–3.62) | 0.2696 | 1.44 | (0.44–4.70) | 0.5441 |
| | | Prominent asymmetry | 0.60 | (0.13–2.87) | 0.5215 | 1.98 | (0.56–7.00) | 0.2849 |
| **Female** | **Unadjusted** | Symmetry | 1 | | | 1.36 | (0.58–3.18) | 0.4763 |
| | | Moderate asymmetry | 1.02 | (0.58–1.81) | 0.9380 | 2.47 | (1.26–4.82) | 0.0087 |
| | | Prominent asymmetry | 1.29 | (0.50–3.31) | 0.5927 | 3.01 | (1.28–7.04) | 0.0116 |
| | **Adjusted** | Symmetry | 1 | | | 1.48 | (0.61–3.60) | 0.3878 |
| | | Moderate asymmetry | 1.12 | (0.60–2.10) | 0.7198 | 2.98 | (1.49–5.94) | 0.0021 |
| | | Prominent asymmetry | 1.37 | (0.49–3.83) | 0.5502 | 3.16 | (1.37–7.29) | 0.0075 |

lower odds of experiencing anxiety than women, they are less sensitive to factors contributing to anxiety.

In this study, women with "low HGS and moderate asymmetry" and "low HGS and prominent asymmetry" had increased odds of having anxiety. Tables 2 and 5 showed that the aOR for anxiety among groups were as follows: 2.173 for the low HGS group, 1.374 for the moderate symmetry group, and 1.829 for the prominent asymmetry group. The former aORs are higher than the latter aORs. Therefore, older women with low HGS and HGS asymmetry may require screening for anxiety.

This study had some limitations. First, the data were insufficient to capture details. For example, there were 542 women with a high school education level or lower, and 108 women with a college education level or higher, representing approximately one-fifth of the former group. Although aOR for the college or higher group (aOR: 3.41, 95% CI: 0.70–16.64) was higher than that of the high school or lower group (aOR: 2.06, 95% CI: 1.22–3.47), there was no statistical significance in the college or higher group, while there was statistical significance in the high school or lower group, according to the subgroup analysis for education level after adjusting for covariates (Table 4). Statistical significance in college or higher group might have been found if there were enough participants in that group. Additionally, we identified more cases with high odds ratios that did not achieve statistical significance, including the rural group (aOR: 2.54, 95% CI: 0.92–6.98), the obese group (aOR: 2.45, 95% CI: 0.94–6.38), and the low-stress group (aOR: 4.83, 95% CI: 0.94–24.73). In addition, we found a positive correlation between the degree of asymmetry and the risk of anxiety both before and after adjusting for covariates (Table 5); however, this effect was non-significant likely because of limited data. We also conducted a subgroup analysis to examine the association between low HGS and anxiety disorder severity. However, as only 21 men and 43 women had moderate or severe anxiety, the range of the 95% CI was too wide, and we were unable to find any significant effects. Therefore, we did not use this table in this study. Second, because a cross-sectional design was used, we could not prove whether there was a causal relationship between low HGS and anxiety. It is possible that low HGS caused anxiety. However, reverse causality and no causality are plausible. Third, some variables used in multiple logistic regression analyses, such as physical activity and smoking, were surveyed through self-reported questionnaires, which were not validated, potentially reducing the accuracy of the presented estimates. Finally, as this study included only South Korean participants, the results may not apply to other countries and contexts.

Measuring HGS is an easy, cost-effective, and noninvasive method for examining sarcopenia in older adults. Few studies have identified an association between low HGS and mental health outcomes. To date, only one study has examined the correlation between low HGS and anxiety. However, the study measured anxiety not by the GAD-7 but by the EuroQol-5D questionnaire and did not examine the correlation between the degree of HGS asymmetry and the risk of anxiety [23]. Anxiety disorders in older adults can result in morbidity and mortality; therefore, it is important that older women with low HGS who are more susceptible to anxiety be monitored.

## Conclusion

This study's findings revealed that low HGS can increase the risk of anxiety in older Korean women. In addition, our findings revealed a positive correlation between the degree of HGS asymmetry and the risks of anxiety in older women, suggesting a need for targeted interventions, which may include public health campaigns promoting exercise to improve HGS. However, we could not examine the association between low HGS and anxiety severity due to the limited number of participants. Further studies with larger sample sizes are required to determine the association between these two factors. In addition, we used Korea's cross-sectional data in this study; further studies are needed to infer causality and identify associations in other ethnic groups.

## Acknowledgments

This study used data from the ninth KNHANES, 2022, KCDC. We would like to thank Editage (www.editage.co.kr) for English language editing.

## Author contributions

**Conceptualization:** Sang-Youn Choi.

**Data curation:** Sang-Youn Choi.

**Formal analysis:** Sang-Youn Choi.

**Investigation:** Sang-Youn Choi.

**Methodology:** Sang-Youn Choi.

**Project administration:** Sang-Youn Choi, Eun-Cheol Park.

**Writing – original draft:** Sang-Youn Choi.

**Writing – review & editing:** Sang-Youn Choi, Su-Min Park, Eun-Cheol Park.

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
