## [Decision Letter · Decision Letter 0]

18 Jan 2025

Dear Dr. Park,

Thank you for submitting your manuscript to PLOS ONE. After careful consideration, we feel that it has merit but does not fully meet PLOS ONE’s publication criteria as it currently stands. Therefore, we invite you to submit a revised version of the manuscript that addresses the points raised during the review process.

We look forward to receiving your revised manuscript.

Kind regards,

Marina De Rui, MD PhD

Academic Editor

PLOS ONE

Journal Requirements:

Reviewers' comments:

Reviewer's Responses to Questions

**Comments to the Author**

1. Is the manuscript technically sound, and do the data support the conclusions?

Reviewer #1: Partly

2. Has the statistical analysis been performed appropriately and rigorously?

Reviewer #1: Yes

3. Have the authors made all data underlying the findings in their manuscript fully available?

Reviewer #1: Yes

4. Is the manuscript presented in an intelligible fashion and written in standard English?

Reviewer #1: Yes

Reviewer #1: This study explores the association between handgrip strength (HGS), HGS asymmetry, and anxiety among older adults in Korea, utilizing data from the 2022 Korea National Health and Nutrition Examination Survey (KNHANES). The finding of a significant correlation between low HGS and anxiety in women is particularly noteworthy. The research addresses an important public health topic and highlights the potential utility of HGS as a marker for mental health issues.

However, the study suffers from critical limitations, particularly the restricted sample size for subgroup analyses, insufficient depth in the interpretation of findings, and a lack of clarity on whether complex sampling analysis was conducted. Additionally, challenges in presenting data effectively further limit the study’s clarity and generalizability.

1. The data source, KNHANES, uses a complex sampling design involving stratification, clustering, and weighting to ensure population representativeness. However, the manuscript does not mention whether these design features were accounted for in the analysis.

A. Clarify whether the complex sampling design was incorporated into the analysis

B. Discuss the implications of not using complex sampling in the limitations section if it cannot be incorporated retroactively.

2. Subgroup analyses (e.g., by education level or area of residence) are undermined by small sample sizes, leading to wide confidence intervals and a lack of statistical significance, even in cases where the effect size appears substantial. For example, in the "college or higher" education subgroup among women, a high odds ratio (OR) was reported, but it did not achieve statistical significance due to the limited sample size.

A. Aggregate or simplify subgroup categories where appropriate to strengthen the robustness of statistical analyses.

B. Discuss the impact of small sample sizes on the interpretation of results more explicitly in the limitations section.

3. The observed gender differences in the results, particularly the lack of significant findings in men, are not sufficiently explained.

A. Elaborate on the biological, social, or cultural factors that may account for the gender differences observed in the association between HGS and anxiety.

B. Include additional references to explain why men might be less affected by HGS changes in relation to anxiety compared to women.

4. The presentation of data in tables can be improved to enhance readability.

A. Including both the odds ratio (OR) and the corresponding p-value in the results table is essential for a more complete and transparent understanding of the findings.

By addressing these points, the study’s rigor and clarity can be improved, enabling a better understanding of the relationship between handgrip strength and anxiety in older adults.

**Do you want your identity to be public for this peer review?** For information about this choice, including consent withdrawal, please see our Privacy Policy

Reviewer #1: No

---

## [Author Response · Author response to Decision Letter 1]

25 Jan 2025

This study explores the association between handgrip strength (HGS), HGS asymmetry, and anxiety among older adults in Korea, utilizing data from the 2022 Korea National Health and Nutrition Examination Survey (KNHANES). The finding of a significant correlation between low HGS and anxiety in women is particularly noteworthy. The research addresses an important public health topic and highlights the potential utility of HGS as a marker for mental health issues.

However, the study suffers from critical limitations, particularly the restricted sample size for subgroup analyses, insufficient depth in the interpretation of findings, and a lack of clarity on whether complex sampling analysis was conducted. Additionally, challenges in presenting data effectively further limit the study’s clarity and generalizability.

-> Thank you for your thorough review of our manuscript. We will carefully read your comments and thoughtfully consider your valuable suggestions. Following the review, we have completely revised our manuscript. Below is our response to each comment.

Point 1

1. The data source, KNHANES, uses a complex sampling design involving stratification, clustering, and weighting to ensure population representativeness. However, the manuscript does not mention whether these design features were accounted for in the analysis.

A. Clarify whether the complex sampling design was incorporated into the analysis

B. Discuss the implications of not using complex sampling in the limitations section if it cannot be incorporated retroactively.

Response 1: We thank for pointing this out. We have added explanation about the complex sampling design incorporated into the analysis.

Revised manuscript, line 126-128, page 7, Method: We used stratification, clustering, and weighting variables in the KNHANES data to ensure population representativeness. We used weighting variables that consider both health and nutrition factors.

Point 2

2. Subgroup analyses (e.g., by education level or area of residence) are undermined by small sample sizes, leading to wide confidence intervals and a lack of statistical significance, even in cases where the effect size appears substantial. For example, in the "college or higher" education subgroup among women, a high odds ratio (OR) was reported, but it did not achieve statistical significance due to the limited sample size.

A. Aggregate or simplify subgroup categories where appropriate to strengthen the robustness of statistical analyses.

B. Discuss the impact of small sample sizes on the interpretation of results more explicitly in the limitations section.

Response 2: We thank you for the comments. Following your suggestions, we attempted to simplify the subgroup categories. However, all the variables were necessary, and we could not aggregate the subgroup categories because each variable had only two categories, except for obesity, which had three. While we initially discussed this limitation in the discussion section, we realized that our explanation was insufficient. Therefore, we have elaborated more explicitly on the limitation of small sample sizes.

Revised manuscript, line 260-263, page 17, Discussion: Additionally, we identified more cases with high odds ratios that did not achieve statistical significance, including the rural group (aOR: 2.54, 95% CI: 0.92–6.98), the obese group (aOR: 2.45, 95% CI: 0.94–6.38), and the low-stress group (aOR: 4.83, 95% CI: 0.94–24.73).

Point 3

3. The observed gender differences in the results, particularly the lack of significant findings in men, are not sufficiently explained.

A. Elaborate on the biological, social, or cultural factors that may account for the gender differences observed in the association between HGS and anxiety.

B. Include additional references to explain why men might be less affected by HGS changes in relation to anxiety compared to women.

Response 3: We thank you for the comments. Initially, we attributed the difference in results to the disparity in average HGS between men and women. However, we found this explanation insufficient and identified the patriarchal mindset as an additional contributing factor.

Revised manuscript, line 230-239, page 16, Discussion: A patriarchal mindset prevalent among the current older Korean adults may also contribute to this difference. Having grown up in a patriarchal society, many of them hold the belief that men should earn money while women should handle housework. As a result, retired men often do not feel obligated to engage in any form of work, including domestic chores, whereas women experience pressure to manage household responsibilities. This dynamic means that women, rather than men, are more likely to feel anxious when low HGS prevents them from fulfilling their expected roles. Given that 49.4% of the male survey population was not working, the disparity between retired men and women may significantly amplify the impact of low HGS on women’s anxiety.

Point 4

4. The presentation of data in tables can be improved to enhance readability.

A. Including both the odds ratio (OR) and the corresponding p-value in the results table is essential for a more complete and transparent understanding of the findings.

Response 4: We thank you for the comments. We have added p-values to all the results tables that were previously missing them.

Revised manuscript, Table 2,3,4,5: You can see the revised tables in 'response to reviewers' file.

---

## [Decision Letter · Decision Letter 1]

6 Feb 2025

Dear Dr. Park,

Thank you for submitting your manuscript to PLOS ONE. After careful consideration, we feel that it has merit but does not fully meet PLOS ONE’s publication criteria as it currently stands. Therefore, we invite you to submit a revised version of the manuscript that addresses the points raised during the review process.

We look forward to receiving your revised manuscript.

Kind regards,

Marina De Rui, MD PhD

Academic Editor

PLOS ONE

Journal Requirements:

Reviewers' comments:

Reviewer's Responses to Questions

**Comments to the Author**

Reviewer #1: (No Response)

2. Is the manuscript technically sound, and do the data support the conclusions?

Reviewer #1: Partly

3. Has the statistical analysis been performed appropriately and rigorously?

Reviewer #1: Yes

4. Have the authors made all data underlying the findings in their manuscript fully available?

Reviewer #1: Yes

5. Is the manuscript presented in an intelligible fashion and written in standard English?

Reviewer #1: Yes

Reviewer #1: This manuscript presents an important study on the association between Handgrip Strength (HGS) and Anxiety. However, there are some concerns regarding the clarity of confounding variables and the lack of Crude OR (Unadjusted OR) in the results tables (Table 2, 3, 4, 5). These issues may reduce the reliability of the findings. To improve the manuscript, the following revisions are strongly recommended.

1. The "Statistical Analysis" section should clearly specify which variables were used as confounders. A detailed list of confounders (e.g., age, education level, marital status, obesity, drinking, sleep duration, stress level, physical activity, smoking, etc.) should be explicitly provided. It should also be clearly stated whether these variables were used only as control variables or analyzed as independent predictors of anxiety.

2. Currently, only Adjusted OR (aOR) is provided, making it difficult to evaluate the effect of confounders on the results. Crude OR (Unadjusted OR) should be included in the tables to allow a direct comparison with Adjusted OR. This will enhance the transparency of how confounding variables affect the observed associations.

3. The current tables (Table 2, 3, 4, 5) provide Adjusted ORs for confounders such as age, education level, marital status, etc.

Confounders are not the primary exposure variables in this study, so providing Adjusted OR for them is not appropriate. The tables should report OR values only for the key independent variables.

Addressing these revisions will improve the clarity and credibility of the study findings.

**Do you want your identity to be public for this peer review?** For information about this choice, including consent withdrawal, please see our Privacy Policy

Reviewer #1: No

---

## [Author Response · Author response to Decision Letter 2]

20 Feb 2025

This manuscript presents an important study on the association between Handgrip Strength (HGS) and Anxiety. However, there are some concerns regarding the clarity of confounding variables and the lack of Crude OR (Unadjusted OR) in the results tables (Table 2, 3, 4, 5). These issues may reduce the reliability of the findings. To improve the manuscript, the following revisions are strongly recommended.

-> We sincerely appreciate the detailed review of our manuscript. We will thoroughly examine your comments and give careful consideration to your insightful suggestions. In response to your feedback, we have made comprehensive revisions to the manuscript. Below, we address each of your comments individually.

Point 1. The "Statistical Analysis" section should clearly specify which variables were used as confounders. A detailed list of confounders (e.g., age, education level, marital status, obesity, drinking, sleep duration, stress level, physical activity, smoking, etc.) should be explicitly provided. It should also be clearly stated whether these variables were used only as control variables or analyzed as independent predictors of anxiety.

Response 1: Thank you for your valuable feedback. We have revised the "Statistical Analysis" section to specify the confounders used in the analysis. As suggested, we have provided a detailed list of confounders, including age, education level, working state, residence, marital status, obesity, drinking, sleep duration, stress level, physical activity, and smoking. Additionally, we have clarified that these variables were included solely as control variables and were not analyzed as independent predictors of anxiety.

Revised manuscript, line 129-131, page 7, Methods: Age, education level, working state, residence, marital status, obesity, drinking, sleep duration, stress level, physical activity, and smoking were used as confounders. These variables were included solely as control variables and not as independent predictors of anxiety.

Point 2 & 3 (related to Table 2,3,4,5)

Point 2. Currently, only Adjusted OR (aOR) is provided, making it difficult to evaluate the effect of confounders on the results. Crude OR (Unadjusted OR) should be included in the tables to allow a direct comparison with Adjusted OR. This will enhance the transparency of how confounding variables affect the observed associations.

Response 2: Thank you for your valuable feedback regarding the presentation of the results. We agree that providing both Crude OR (Unadjusted OR, cOR) and Adjusted OR (aOR) in the tables would enhance transparency and allow for a clearer evaluation of the effect of confounders on the observed associations.

In response to your comment, we have revised Tables 2, 3, 4, and 5 to include both Crude OR and Adjusted OR. This modification allows for a direct comparison between unadjusted and adjusted estimates, providing a more comprehensive understanding of how confounding variables influence the results. We also added explanations in manuscript to clarity if the results were adjusted for covariates or not.

After appending the Crude ORs to Table 3, the table became excessively large. To improve readability and organization, we divided the content into two separate tables: Table 3 and Table 4. Subsequently, all following tables were renumbered accordingly.

We believe these changes improve the clarity and robustness of our findings. Thank you for your constructive suggestion, which has significantly strengthened the manuscript.

Point 3. The current tables (Table 2, 3, 4, 5) provide Adjusted ORs for confounders such as age, education level, marital status, etc.

Confounders are not the primary exposure variables in this study, so providing Adjusted OR for them is not appropriate. The tables should report OR values only for the key independent variables.

Response 3: Thank you for your insightful comment regarding the presentation of Adjusted Odds Ratios (ORs) for confounders in the tables. We agree that confounders such as age, education level, marital status, etc., are not the primary exposure variables in this study, and reporting Adjusted ORs for these variables may not be appropriate.

In response to your feedback, we have revised Table 2 to remove the Adjusted ORs for confounders. Also, we have deleted the accompanying sentences in the text that explained the Adjusted ORs for confounders. The tables now focus solely on reporting aOR values for the key independent variables, as recommended.

We divided the participants by each confounder for Table 3 and 4 (previously Table 3). We conducted multiple logistic regression analysis between HGS and anxiety in each subgroup. Since these subgroup analyses provided meaningful insights, we have retained them in the manuscript.

In Tables 5 and 6 (previously Tables 4 and 5), ORs for confounders were not provided because the confounders were used exclusively to adjust for covariates. Therefore, we have retained Tables 5 and 6 in the manuscript.

We believe these changes improve the clarity and relevance of the results presented. Thank you for your valuable suggestion, which has helped enhance the quality of our manuscript.

Revised manuscript, line 162-164, page 10, Results: Both before and after adjusting for covariates, the risk of having anxiety was significantly higher among women who had low HGS (crude OR [cOR]: 1.33, 95% CI: 0.67–2.63 in men, cOR: 1.989, 95% CI: 1.23–3.21 in women, adjusted OR [aOR]: 1.22, 95% CI: 0.62–2.43 in men, aOR: 2.17, 95% CI: 1.31–3.61 in women).

Revised manuscript, line 172-175, page 11, Results: Both before and after adjusting for covariates, no statistically significant differences were found in any male subgroup, in contrast to female subgroups.

Revised manuscript, line 261-265, page 18, Discussion: Although aOR for the college or higher group (aOR: 3.41, 95% CI: 0.70–16.64) was higher than that of the high school or lower group (aOR: 2.06, 95% CI: 1.22–3.47), there was no statistical significance in the college or higher group, while there was statistical significance in the high school or lower group, according to the subgroup analysis for education level after adjusting for covariates (Table 4).

Revised manuscript, line 269-271, page 18, Discussion: In addition, we found a positive correlation between the degree of asymmetry and the risk of anxiety both before and after adjusting for covariates (Table 5);

Revised manuscript, Table 2,3,4,5,6 : Please refer to "Respond to Reviewers" file or manuscript file.

---

## [Decision Letter · Decision Letter 2]

3 Mar 2025

Association between handgrip strength, handgrip strength asymmetry, and anxiety in Korean older adults: The Korean National Health and Nutrition Examination Survey 2022

PONE-D-24-53685R2

Dear Dr. Park,

We’re pleased to inform you that your manuscript has been judged scientifically suitable for publication and will be formally accepted for publication once it meets all outstanding technical requirements.

Kind regards,

Marina De Rui, MD PhD

Academic Editor

PLOS ONE

Additional Editor Comments (optional):

Reviewers' comments:

Reviewer's Responses to Questions

**Comments to the Author**

Reviewer #1: All comments have been addressed

2. Is the manuscript technically sound, and do the data support the conclusions?

Reviewer #1: Yes

3. Has the statistical analysis been performed appropriately and rigorously?

Reviewer #1: Yes

4. Have the authors made all data underlying the findings in their manuscript fully available?

Reviewer #1: Yes

5. Is the manuscript presented in an intelligible fashion and written in standard English?

Reviewer #1: Yes

Reviewer #1: Tables 2 to 5, both unadjusted (crude) and adjusted odds ratios are presented, which is commendable for transparency. However, for ease of interpretation, it is recommended that the order of the estimates be reversed so that the unadjusted (crude) odds ratios are listed first, followed by the adjusted odds ratios. Presenting the crude values first will clearly delineate the impact of confounding variables and allow readers to directly compare the effect of adjustment on the associations.

**Do you want your identity to be public for this peer review?** For information about this choice, including consent withdrawal, please see our Privacy Policy

Reviewer #1: No

---

## [Editor Report · Acceptance letter]

PONE-D-24-53685R2

PLOS ONE

Dear Dr. Park,

I'm pleased to inform you that your manuscript has been deemed suitable for publication in PLOS ONE. Congratulations! Your manuscript is now being handed over to our production team.

Kind regards,

on behalf of

Dr. Marina De Rui

Academic Editor

PLOS ONE